# Sustainability of Higher Education: Study of Student Opinions about the Possibility of Replacing Teachers with AI Technologies

Valery Okulich-Kazarin [1], Artem Artyukhov [2,3], Łukasz Skowron [4], Nadiia Artyukhova [3], Oleksandr Dluhopolskyi [5,6,*] and Wiktor Cwynar [6]

1 Faculty of Social Sciences and Humanities, Humanitas University, 41-200 Sosnowiec, Poland; okwalery@gmail.com
2 Faculty of Commerce, University of Economics in Bratislava, 852-35 Bratislava, Slovakia; a.artyukhov@pohnp.sumdu.edu.ua
3 Academic and Research Institute of Business, Economics and Management, Sumy State University, 40-007 Sumy, Ukraine
4 Faculty of Management, Lublin University of Technology, 20-618 Lublin, Poland
5 Faculty of Economics and Management, West Ukrainian National University, 46-027 Ternopil, Ukraine
6 Institute of Public Administration and Business, WSEI University, 20-209 Lublin, Poland
* Correspondence: dlugopolsky77@gmail.com

**Abstract:** The rapid development of artificial intelligence (AI) has affected higher education. Students now receive new tools that optimize the performance of current tasks. Universities have also begun implementing AI technologies to help university teachers and improve the quality of educational services and solve the Sustainable Development Goal 4. Hypothetically, it is possible to replace university teachers by using AI technologies. This is a hidden conflict of Sustainable Development Goal 4 and Sustainable Development Goal 8. This research aimed to examine the perceptions of Eastern European students about the possibility of replacing university teachers through AI technologies. The authors used an information study with a bibliometric analysis of 2000 sources, planning the experiments and compiling the questionnaire, surveying 599 students using an electronic questionnaire and cloud technologies, statistical processing questionnaires using Excel tables, and verifying statistical hypotheses. Verification of statistical hypotheses for replies of 599 respondents showed that more than 10% of the surveyed students from Eastern European universities are confident that AI will replace university teachers in five years. It was shown that the opinions of students in the 1st stage (undergraduate study) from the countries of the European Union and countries outside the European Union have significant differences. The obtained results were proven using one-sided testing and standard hypothesis testing level, $\alpha = 0.05$. The article was completed with multilevel managerial and pedagogical recommendations. These recommendations are designed to increase higher education's sustainability in AI implementation.

**Keywords:** sustainability; Sustainable Development Goal (SDG); SDG 4; SDG 8; artificial intelligence; higher education; students; university teachers

## 1. Introduction

The rapid development of artificial intelligence (AI) has affected many branches of science and technology and has also caused a change in approaches to the educational process. Students and teachers have new tools that make their work easier and optimize educational tasks. At the same time, AI possibilities can replace a teacher's work and a student's educational activity in completing assignments.

In this article, the authors published the results of information, experimental and computational studies regarding the sustainability of higher education using AI technologies. The use of AI may threaten the sustainability of higher education.

The use of any tools and approaches in education is directly related to students' feedback on the effectiveness of their actions and the prospects for further implementation [1–3]. In this way, data on the model of a sustainable university are accumulated, and the challenges facing education are analyzed.

For example, in [1], authors from Romania examined students' opinions about well-being and grades in a sustainable university. Using research results can lead to a more supportive and motivating learning environment, which can lead to improved academic performance and mental health.

The authors [2] studied Brazil's efficiency factors and sustainability indicators of higher education management. The results obtained by the authors show that competency management directly impacts work processes. The focus on the student and society directly impacts strategic planning. It follows from this fact that the opinion of students becomes very important for ensuring the sustainability of university management.

Authors from China [3] studied the tools for training university teachers for the further sustainable development of higher education institutions. They linked the systemic growth of teaching staff to the sustainable development of higher education.

AI has proclaimed a cybernetic panacea to improve and expand educational services [4]. In the US, the Georgia Institute of Technology, professors have used virtual teaching assistants for several years [5]. They report that teachers and students are satisfied with the practical result [5]. Today, AI can measure how well students have mastered knowledge and adjust educational content accordingly [4]. The AI can give students fair grades [4], eliminating the problems found by Romanian authors [1]. In connection with the above, hypothetically, AI can replace university teachers.

The European Union forecasts AI will radically change the education system [4]. In [4], AI is shown as a tool for solving several educational problems:

- AI will solve the problem of teacher workload; this solution will solve the problem of retaining qualified specialists
- CenturyTech's AI-powered product promises to "strengthen" teaching and solve the problem of a "one size fits all" education model by providing students with personalized learning
- Personalized learning systems promise to reduce the achievement gap
- AI can remove barriers to social mobility (however, under certain circumstances, AI can lead to inequality).

Therefore, while developing the research described in the article [3], society should know whether there is a threat of a complete replacement of university teachers by AI technologies. Based on the results of a literature review, including source [2], the authors of this study studied the opinions of 599 students to ensure the sustainability of higher education.

The purpose of the work was to examine the perceptions of Eastern European students about the possibility of replacing university teachers using AI technologies. The authors studied the students' perceptions concerning the future period equal to five years (2023–2027).

The practical value lies in new empirical data and new research methodology. The need for further empirical evidence is the opinion of Eastern European students about the possibility of AI to replace teachers in the higher education system. Two kinds of conflicts in the higher education system were described based on sustainable development goals. A modern methodology for researching this relatively new technological phenomenon was also created and tested. Using the latest methodology, the authors, for the first time, received and analyzed the opinions of Eastern European students on the topic: will AI replace university teachers in five years? Verification of statistical hypotheses transformed the subjective opinions of 599 students into new scientific knowledge.

New data are the basis for monitoring students' opinions on a given topic in the near future. Based on monitoring, it is possible to create a mathematical model that describes the rate of spread of AI in higher education.

The importance and usefulness of the study are emphasized by the fact that the United Nations pays close attention to the topic of AI. On 10/26/2023 the United Nations published a message about creating the "AI Advisory Body." Mr. Guterres pointed out, "It can be stimulating. . . efforts to achieve the 17 Sustainable Development Goals (SDGs) by 2030" (https://news.un.org/en/story/2023/10/1142867, accessed on 20 October 2023). Among other tasks of the "AI Advisory Body" is the task of resolving crises in education services.

The authors explored the following two divergent hypotheses.

**Hypothesis 1.** *All of the students are sure AI will replace university teachers.*

**Hypothesis 2.** *There are no students who are sure that AI will replace university teachers.*

The obtained results contribute to a better scientific understanding and forecasting of global changes in the market of pedagogical work in higher education.

Information research, bibliometric analysis, student survey, and verification of statistical hypotheses led to the following main conclusions:

- Introducing AI in higher education may conflict between two global sustainable development goals (SDG). These are SDG 4 and SDG 8
- More than 10% of surveyed students from Eastern European countries are sure that AI will replace university teachers in five years
- There is a difference in the opinions of students in the 1st stage (undergraduate study) from the countries of the European Union and countries outside the European Union. The share of students who are confident that AI will replace university teachers is higher in Kazakhstan and Ukraine. In Poland and Slovakia, the percentage of such students is smaller.
- Multilevel managerial and pedagogical recommendations formulated by the authors can increase the sustainability of higher education in implementing AI. These recommendations are for university leaders (SDG 4), governments and politicians (SDG 8), researchers (SDG 8), and university teachers (SDG 4). These recommendations are not for AI developers.

Validity, relativity, and reliability of the results were ensured as follows: methodological validity of the initial positions; using research methods that are adequate for their purpose, objectives, logic, and scope; using a reliable system, clear instructions, and a simple user interface; lack of influence of the observer on the observed; professionalism and scientific rigor; representativeness and statistical significance of experimental data; impartiality of assessment during their processing and interpretation; consistency of conclusions.

## 2. Literary Review

### 2.1. Theoretical Framework

The definition of "sustainability" is interpreted differently depending on time and context [6]. According to [7], you may define "sustainability" as "maintaining well-being over a long, perhaps even an indefinite period." At the same time, there is an official definition of the United Nations "Sustainable development is how we must live today if we want a better tomorrow, by meeting present needs without compromising the chances of future generations to meet their needs" (https://www.un.org/sustainabledevelopment/blog/2023/08/what-is-sustainable-development/, accessed on 20 October 2023). By following the official definition and meeting the current needs of students (SDG 4) using AI tools, society, as well as higher education leaders and policymakers, must ensure that future generations of university teachers meet their needs for decent work (SDG 8).

Sustainable education is an approach to learning and teaching that focuses on the environmental, economic, and social sustainability of our planet [1]. Sustainable education encourages students to understand how the decisions they make today can impact the

world tomorrow [1]. Our study examined the essential question asked of students about the consequences that may occur in 5 years, that is, in 2027.

AI in education is not only a tool for performing work and obtaining a "product" according to a given technical task. This is an element of modern digital learning [8–10], which is an essential component of various educational platforms [11–13] that form the necessary skills of graduates [14,15]. An important "bridge" between the university and the labor market is new teaching methods in the context of sustainable development goals [16,17] and the economic growth of regions. The internal (university) education quality assurance system has an effective way of improvement through stakeholder surveys [18], including the role of AI in the learning process.

Researchers often write that sustainability has three components: environmental, economic, and social [19]. In some publications, the authors argue that the environmental component is the most important [20–22]. The economic component is a smaller subset of the social component [20–22].

Our study covers at least two components of sustainability.

The authors put the social component in first place. This study addresses Goal 4 of the 17 global sustainable development goals (SDG) [23]. This goal is "Ensure inclusive and equitable quality education and encourage lifelong learning opportunities for all" [24]. In detail, these are goals 4.3, 4.4, and 4C [24].

To achieve SDG 4, measures such as increasing the number of teachers, improving basic school infrastructure, and implementing digital transformation are extremely important [25]. It looks natural that the introduction of new pedagogical AI tools leads to an increase in the quality of educational services [8–10] and a good step to achieve SDG 4 [25].

In connection with the above [2–5], hypothetically, AI can replace university teachers in the future.

The displacement of university teachers from the labor market into the sector of the unemployed conflicts with the economic component of sustainability (SDG 8). An important "bridge" between the university and the labor market is new teaching methods in the context of sustainable development goals [16,17].

In this aspect, it is the SDG 8 of the 17 global sustainable development goals [23]. This goal is "promotion of steady, inclusive and sustainable economic growth, full and productive employment and decent work for all" [23]. In detail, there is goal 8.8 [26].

If AI forces university teachers out of the knowledge labor market, they will have to look for work in other, non-intellectual areas of labor activity.

It is necessary to know how society will benefit from an increase in the quality of education using AI technologies. This fact must be weighed against the threat to the knowledge economy if the most skilled workers leave the labor market. In this case, the hidden conflict between SDGs 4 and 8 is external, concerning the higher education system. After all, the displacement of university teachers from universities will eventually lead to a decrease in the growth rates of the regional economy.

The uncontrolled intervention of AI technologies in higher education can turn the hidden conflict of SDGs 4 and 8 into an internal conflict in higher education. Even though the internal (university) quality assurance system of education has an effective way to improve through stakeholder surveys [18], the mass dismissal of university teachers will decrease the creative and educational potential of the higher education system.

Studying students' opinions about the possibility of replacing university teachers with AI technologies is one of the first steps towards resolving the hidden conflict between SDGs 4 and 8.

### 2.2. Bibliometric Analysis

Considering the dataset of articles, a literature review on AI in education was carried out using bibliometric analysis tools in several stages. Each stage made it possible to "cut off" from the general array of publications on AI those sources which do not relate to the subject of this work. When conducting a bibliometric analysis, it was also considered

that AI in this work is considered not from a technical, but from an ideological point of view, the prospects for its implementation in the educational process. The specialization of the interviewed applicants for higher education was also considered when choosing the branches of knowledge to which certain literary sources belonged.

During the first stage, from the general array of publications from the Scopus database (https://www.scopus.com/, accessed on 20 October 2023), the authors separated publications for 2018–2023 (220,633 articles). The second stage of sorting consisted of selecting areas of knowledge where AI is studied not as a technical tool but as a phenomenon with subsequent behavioral, social, and economic reactions to its use. Thus, after this sorting stage (fields of knowledge were selected: Decision Sciences; Social Sciences; Business, Management and Accounting; Economics, Econometrics and Finance), 41,188 articles were received for further bibliometric analysis.

During the third stage, the 2000 most cited articles were selected (according to https://www.scopus.com/), which became the subject of bibliometric analysis.

Based on bibliometric analysis (analysis tool—VOSviewer, version 1.6.19, https://www.vosviewer.com/, open access software), 12,383 keywords were obtained from which 325 keywords were selected (at least ten mentions in the array of articles). The list of keywords excluded those that did not belong to the field of research and did not have a scientific component. Based on these keywords, a map was built (Figure 1).

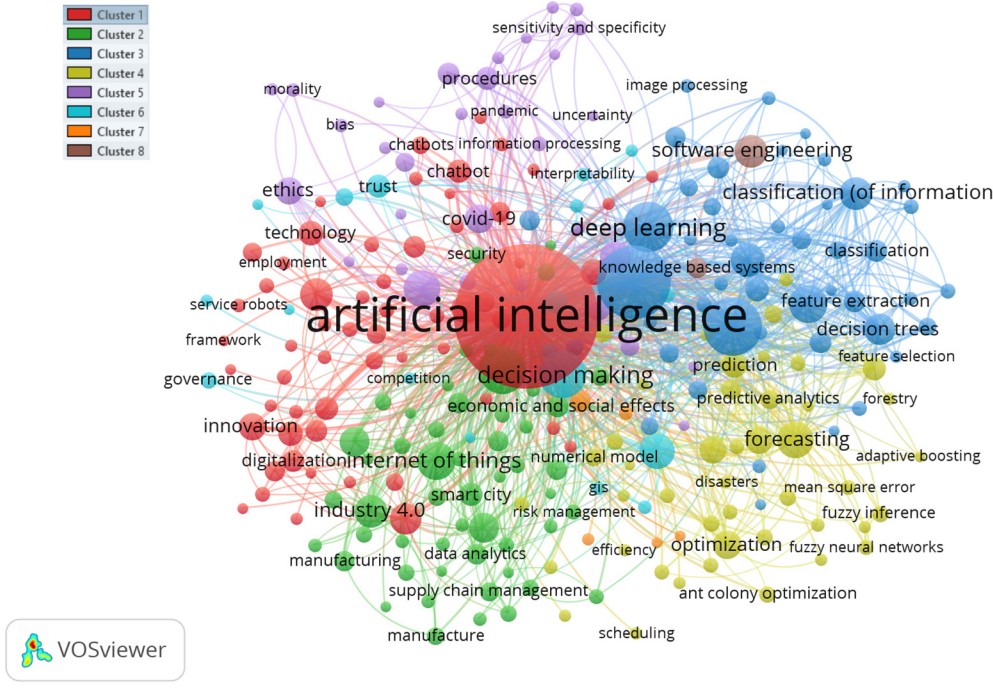

**Figure 1.** Keyword map for the query "artificial intelligence" (dataset of articles for analysis—https://www.scopus.com/, analysis tool—VOSviewer).

In the keyword map for the query "artificial intelligence", we found interesting clusters (Figure 1).

The largest cluster (cluster 1) connects AI with education (keywords are "active learning", "e-learning", "education", "higher education", "knowledge", and "learning") and, what is very important, participants of the educational process ("students") and their opinion ("surveys").

Cluster 2, the second largest, links AI with socio-economic development (economic and social effects, industry 4.0, sustainable development).

It is also important that in clusters 1 and 2, there is a connection between AI and the quality assessment process ("service quality", "quality control").

The importance of obtaining data on AI in education is also confirmed by the popularity of the educational vector in the body of literature devoted to all areas of AI study. As the data in Figure 2 (embedded in https://www.scopus.com/ tool) shows, the top 1% of topics by prominence include various types of learning, which shows the "depth" of penetration of AI into the educational field.

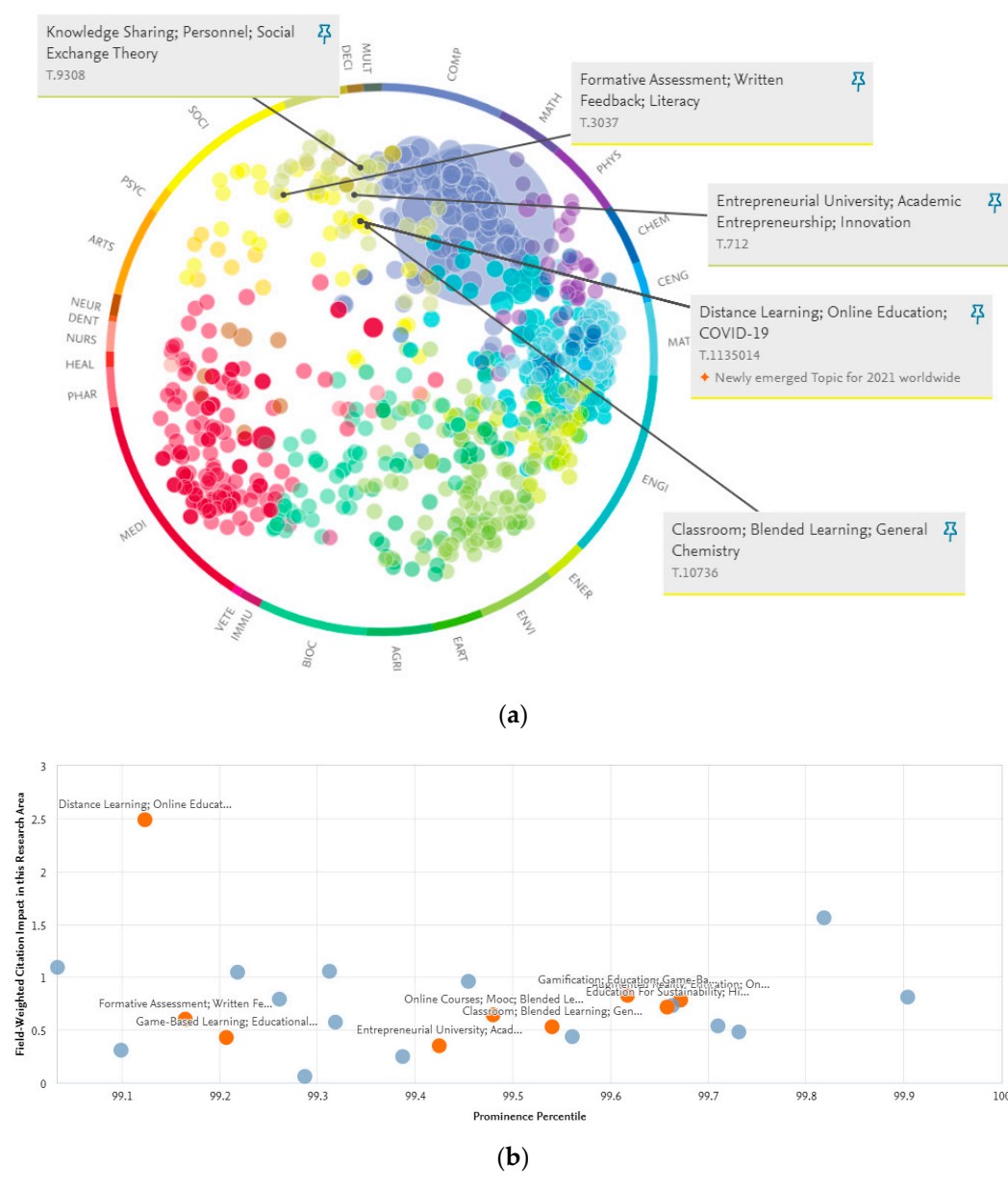

(**a**)

(**b**)

**Figure 2.** Wheel (**a**) and sifting (**b**) of topics for the query "artificial intelligence", top 1% of topics by prominence (dataset of articles for analysis—https://www.scopus.com/, analysis tool—SciVal).

Reviews devoted to AI in education [27–38] focus on the evolution, implementation, and improvement of approaches and tools, not on assessing the prospects and challenges that education faces in the era of total artificial "intellectualization" (Figure 2). In late 2022 to early 2023, with the open access to ChatGPT, several discussions about academic integrity when using AI (for example [39,40]) as well as the first recommendations for the education sector [41] appeared.

All the above facts testify to the relevance of this study and the need to survey students about the prospects for AI in education.

## 3. Materials and Methods

### 3.1. General Information

The study was carried out from December 2022 to August 2023 at National Louis University (Poland), Mieszko I University of Applied Sciences in Poznan (Poland), Karaganda University named after Academician Buketov (Kazakhstan), University of Economics in Bratislava (Slovakia), West Ukrainian National University (Ukraine), Dnipro University of Technology (Ukraine).

The authors used modern standard research methods to study students' opinions about the possibility of replacing university teachers with AI. These were such reliable and economical methods as shown below:

Information research with a step-by-step application of bibliometric analysis tools

- Planning an experiment and compiling a questionnaire
- Survey of respondents using cloud technologies (an electronic form of the questionnaire)
- Primary processing of experimental results (visual presentation in the form of tables and diagrams)
- Statistical processing of questionnaires using Excel spreadsheets and verification of statistical hypotheses.

Two thousand most cited articles (according to https://www.scopus.com/) were used for bibliometric analysis.

The authors explored two divergent hypotheses.

Hypothesis 1 was converted into two statistical hypotheses [42].

Research hypothesis 1 claims that all students are sure that AI will replace university teachers in five years if random deviations are not considered.

The Research hypothesis 1 is written: $\mu_0 = 100.00\%$.

Alternative hypothesis 1 claims that not all students are sure that AI will replace university teachers in five years if random deviations are not considered.

The Alternative hypothesis 1 is written: $\mu_0 < 100.00\%$.

Hypothesis 2 also was converted into two statistical hypotheses [42].

Research hypothesis 2 claims that no student is sure that AI will replace university teachers in five years if random deviations are not considered.

The Research hypothesis 2 is written: $\mu_0 = 0.00\%$.

Alternative hypothesis 2 claims that some students are sure that AI will replace university teachers in five years if random deviations are not considered.

The Alternative hypothesis 2 is written: $\mu_0 > 0.00\%$.

A comprehensive research methodology includes the step-by-step use of the following research tools: (a) bibliometric analysis (analysis tool—VOSviewer, https://www.vosviewer.com, open access software, SciVal, https://www.scival.com, accessed on 20 October 2023, Scopus database add-on for bibliometric analysis), in addition to the use of a standard literature review, (b) the use of an electronic questionnaire to survey instead of a traditional paper questionnaire, (c) the use of cloud technologies to post the questionnaire in the public domain, (d) the use of AI tools for preliminary processing of measurement results.

The basis for the survey of students at Eastern European universities is the weakness of the Eastern European market for higher educational services. The volume of the educational services market in Eastern Europe is 6.1% of the total volume of the Eurasian market of educational services [43]. Compared to the Western European market, this is about ten times less. In addition, the authors of the article [43] statistically proved that the publication activity on the problems of higher education in Eastern European countries is low if random deviations are not considered. The countries were chosen in such a way as to ensure maximum diversity. Therefore, two EU countries (Poland and Slovakia) and two non-EU countries (Kazakhstan and Ukraine) were selected. The selected countries provide the maximum geographical and religious-cultural diversity.

### 3.2. Experiment Planning

When planning the experiment, the authors relied on the results of previous studies in different countries worldwide. The authors followed the next steps when planning the experiments [44]. To reduce the number of experiments while maintaining the high quality of the study, the authors planned and performed the following:

- Determination of the purpose of the experiment
- Selecting countries to compare results
- Identification of all educational levels and experimental units (groups) of respondents
- Selecting the measurements to be performed (a list of 12 essential questions) and the measurement method
- Selection of rules for obtaining experimental data (5 options for answers to essential questions)
- Conducting a pilot experiment to assess validity, reliability, and relativity (not described in this article)
- Calculation of the number of observations that need to be carried out

The matrix of the experiment plan is shown in Table 1.

**Table 1.** The matrix of the experiment plan.

| Country | Undergraduate Study | Master Study | Postgraduate Study | Master of Business Administration |
|---------|---------------------|--------------|--------------------|-----------------------------------|
| Kazakhstan | + | | | |
| Poland | + | + | + | + |
| Slovakia | + | | | |
| Ukraine | + | + | | |

Table 1 shows that eight groups of respondents were scheduled for the survey. The first comparative line consists of four groups of students from different countries in the 1st stage (undergraduate study) (vertical line in Table 1). The second comparative line consists of four groups of respondents from Poland (horizontal line in Table 1). These are students of the 1st and 2nd stages, students of regular postgraduate courses, and students of the Master of Business Administration (MBA) Program. The MBA program is a training program for preparing the most qualified leaders. This fact distinguishes the MBA program from other postgraduate programs.

All respondents studied social sciences, humanities, and engineering unrelated to IT technologies. This means that the respondents did not professionally study subjects related to AI.

The complete experiment plan includes a survey of 16 groups of respondents (Table 1). However, evidence-based planning helped to reduce the number of experimental groups to 8. At the same time, the authors retained the ability to compare students' opinions by country (different cultures) and educational level (Table 1). Science-based planning of the experiments reduced the cost of time, human, and financial resources by two times.

### 3.3. General Description of the Questionnaire

The questionnaire was compiled following the standard requirements for conducting surveys and their processing [44]. The questionnaire includes an appeal to respondents, a metric, and a main part.

In an address to the respondents, the authors informed the students that participation in the survey was voluntary and anonymous.

The metric part includes four general questions: Level of study, Age, Gender, and Country. This is the necessary and sufficient information to identify groups of respondents.

The main part includes twelve essential questions. These questions are about students' opinions about various aspects of the application of AI in higher education.

In this article, the authors drew attention to only one essential question. This question is as follows: Will AI replace university teachers in five years?

Respondents could choose one of five answers:

- definitely yes;
- rather yes;
- have no idea;
- rather not;
- definitely no.

In further calculations, the first two answers were summed up and considered as students' confidence that AI will replace university teachers within five years.

The last two answers were also summed up. They were viewed as students confident that AI would not replace university teachers within five years.

The middle answer reflects students' doubts and lack of confidence that AI will replace university teachers within five years.

Primary processing of the experiment results included their visual presentation in the form of tables and diagrams.

### 3.4. General Description of the Respondent Groups

Data were collected from 599 public and non-public sector university students from four countries. A serial (nested) sample was used for the experiment. Serial (nested) sampling assumes that series or groups of population units should be selected [44]. There were eight groups of respondents in total.

When choosing the surveyed groups, the authors sought maximum diversity.

The experimental conditions must be very varied for conclusions to be broad in scope. However, a negative consequence of increasing the size of the experiment is the increase in response variability. Blocking is a technique that can often be used to help manage this problem [44]. To block an investigation means to divide observations into groups, called blocks, so that the observations in each block are collected under relatively similar experimental conditions. The authors used a blocking method to achieve maximum diversity while maintaining the required accuracy.

For each group of respondents, the authors prepared a separate questionnaire. Each questionnaire was hosted in the cloud service of the National Louis University. This measure eliminated errors in obtaining student opinions and their subsequent processing. All ethical principles were observed during the survey.

General information about the respondents is presented in Table 2.

**Table 2.** General information about respondents.

| Country | University | Level of Study | Male/Female/ Other | Number of Respondents |
|---|---|---|---|---|
| 1. Kazakhstan | Karaganda University named after Academician Buketov | Undergraduate | 29/43/1 | 73 |
| 2. Poland | Mieszko I University of Applied Sciences in Poznan | Undergraduate | 39/17/0 | 56 |
| 3. Poland | National Louis University | Master | 13/26/0 | 39 |
| 4. Poland | National Louis University | Postgraduate | 13/60/0 | 73 |
| 5. Poland | National Louis University | Master of Business Administration | 36/24/0 | 60 |
| 6. Slovakia | University of Economics in Bratislava | Undergraduate | 33/27/0 | 60 |
| 7. Ukraine | West Ukrainian National University | Undergraduate | 31/86/1 | 118 |
| 8. Ukraine | Dnipro University of Technology | Undergraduate | 84/36/0 | 120 |
| Total | - | - | 278/319/2 | 599 |

Table 2 shows that eight groups of respondents from four Eastern European countries were interviewed in the empirical part of the study. The minimum number of respondents in a group is 39. The maximum number of respondents in a group is 120. The total number of respondents interviewed is 599. The study involved full-time students, both male (278) and female (319). There were also 2 participants of a different gender orientation. The age of the participants ranged from 18 to 64 years.

The groups selected for the survey (Table 2) fully corresponded to the plan of the experiment (Table 1).

The authors studied students' opinions for the first time to get the most general picture. Therefore, Table 2 does not show the gender composition of the respondents.

### 3.5. Method of Verification of Statistical Hypotheses

The method of hypothesis testing about the unknown average is to calculate t-statistics [42,45]. Verification of statistical hypotheses is based on comparing the average of the sample, $M_{(x)}$, with a given number $\mu_0$ [42].

It should be kept in mind that this study has two divergent hypotheses. Both divergent hypotheses were transformed into statistical hypotheses [42].

A one-sided test was chosen because the number of students sure that AI will replace university teachers in five years cannot exceed 100.00%.

A one-sided test was chosen because the number of students who are not sure that AI will replace university teachers in five years cannot be less than 0.00%.

In both cases, the authors applied the standard hypothesis testing level, $\alpha = 0.05$ [42].

The purpose of the study was divided into three narrow tasks: (a) to verify hypotheses 1 and 2 for all respondents; (b) to verify hypothesis 1 for individual groups of respondents; (c) to verify hypothesis 2 for individual groups of respondents.

After verifying the statistical hypotheses and discussing the results, the authors summarized the study results and prepared recommendations.

## 4. Results

The section describes the empirical results, their comparison and interpretation, and some conclusions that should be shown.

### 4.1. Primary Results of the Experiment (Students' Survey)

The respondents' answers are summarized in Table 3. Here N is the number of respondents.

**Table 3.** Distribution of respondents' answers.

| Group of Respondents | N | Definitely Yes | Rather Yes | I Have No Idea | Rather Not | Definitely No |
|---|---|---|---|---|---|---|
| 1. Kazakhstan | 73 | 7 | 16 | 19 | 24 | 7 |
| 2. Poland | 56 | 0 | 1 | 17 | 26 | 12 |
| 3. Poland | 39 | 2 | 2 | 5 | 14 | 16 |
| 4. Poland | 73 | 0 | 2 | 17 | 35 | 19 |
| 5. Poland | 60 | 0 | 2 | 13 | 21 | 24 |
| 6. Slovakia | 60 | 0 | 1 | 14 | 38 | 7 |
| 7. Ukraine | 118 | 8 | 10 | 21 | 32 | 47 |
| 8. Ukraine | 120 | 5 | 9 | 18 | 44 | 44 |
| Total | 599 | 22 | 43 | 124 | 234 | 176 |

Table 3 states that some students are confident that AI will replace university teachers within five years.

Recall that the answers of the respondents were grouped into three groups:

- students' confidence that AI will replace university teachers within five years;
- students' confidence that AI will not replace university teachers within five years;

- students doubt and lack confidence that AI will replace university teachers within five years.

Respondents from the first group were confident that AI would replace university teachers within five years. These respondents chose the answer options "Definitely yes" and "Rather yes". Their number was 65. The rest of the students were unsure or doubted that AI would replace university teachers within five years.

Therefore, the authors worked mainly with respondents from the first group.

The distribution of responses for the total number of respondents is shown in Figure 3. Figure 3 clearly shows that respondents from group 1 are in the minority. There are only 65 of them. The rest of the students are not sure that AI will replace university teachers within five years. And on the contrary, most of them are convinced that AI will not replace university teachers within five years.

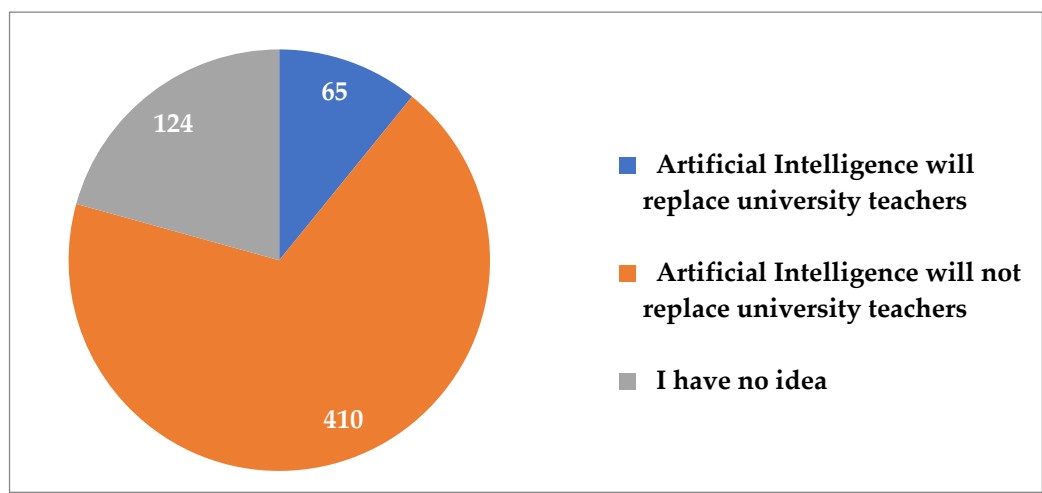

**Figure 3.** Student opinions: Will AI replace university teachers?

This number (65) may be the result of random deviations. Also, this number (65) may result from a combination of respondents' subjective opinions.

Unfortunately, Table 3 and Figure 3 do not give a clear answer: Can they consider that the responses of 65 respondents are the result of random deviations?

In other words, it is not proven that the answers of 65 respondents can be ignored. Therefore, no one can reject Hypothesis 1 and accept Hypothesis 2.

Verification of statistical hypotheses will show a clear, scientifically sound answer.

*4.2. Verification of Hypotheses 1 and 2 for All Respondents*

Here we find out if all students are sure that AI will replace university teachers in five years?

Statistical indicators for testing statistical hypotheses are shown in Table 4.

**Table 4.** Statistical indicators of all respondents' answers.

| Group of Respondents | N | $M_{(x)}$ | $\delta_x$ | $\delta_{x-1}$ |
|---|---|---|---|---|
| Total | 599 | 10.85 | 31.10 | 31.13 |

Table 4 shows the statistical indicators that are needed to verify statistical hypotheses. These are sample size (N), average of the sample ($M_{(x)}$), standard deviation for sample ($\delta_x$), and standard deviation for population ($\delta_{x-1}$).

We start by testing hypothesis 1.

Verification of statistical hypotheses for hypothesis 1. was performed for the total number of respondents in Table 5.

**Table 5.** Verification of hypothesis 1 for all respondents (all students are sure that AI will replace university teachers, $\mu_0 = 100.00$).

| Statistical Indicators | Value |
|---|---|
| Sample size, $N$ | 599 |
| Average of the sample, $M_{(x)}$ | 10.85 |
| Standard deviation for sample, $\delta_x$ | 31.10 |
| Average error, $\dot{S}_{\dot{X}} = \delta_x / \sqrt{n}$ | 1.27 |
| Value $\mid t_{stat} \mid$ for $\mu_0 = 100.00\%$, $(M_{(x)} - \mu_0)/\dot{S}_{\dot{X}}$ | 70.20 |
| Value $t_{tabl}$ for a standard testing level of $\alpha$ *(0.05)* | 1.645 |
| $\mid t_{stat} \mid > t_{tabl}$ | Yes |

In Table 5, t-statistics $\mid t_{stat} \mid$ is larger than the $t_{tabl}$ for the given number (100.00%). So, the authors have accepted Alternative Hypothesis 1: Not all students are sure that AI will replace university teachers in five years if random deviations are not considered. The result was obtained with a standard hypothesis testing level that equals 0.05.

Let us go back to Hypothesis 2.

The statistical hypotheses for hypothesis 2 were verified for the total number of respondents in Table 6.

**Table 6.** Verification of hypothesis 2 for all respondents (there are no students who are sure that AI will replace university teachers, $\mu_0 = 0.00$).

| Statistical Indicators | Value |
|---|---|
| Sample size, $N$ | 599 |
| Average of the sample, $M_{(x)}$ | 10.85 |
| Standard deviation for sample, $\delta_x$ | 31.10 |
| Average error, $\dot{S}_{\dot{X}} = \delta_x / \sqrt{n}$ | 1.27 |
| Value $\mid t_{stat} \mid$ for $\mu_0 = 0.00\%$, $(M_{(x)} - \mu_0)/\dot{S}_{\dot{X}}$ | 8.54 |
| Value $t_{tabl}$ for a standard testing level of $\alpha$ *(0.05)* | 1.645 |
| $\mid t_{stat} \mid > t_{tabl}$ | Yes |

In Table 6, t-statistics $\mid t_{stat} \mid$ is larger than the $t_{tabl}$ for the given number (0.00%). So, the authors have accepted Alternative Hypothesis 2: some students are sure that AI will replace university teachers in five years if random deviations are not considered. The result was obtained with a standard hypothesis testing level that equals 0.05.

Verification of statistical hypotheses showed that 65 respondents who are sure that AI will replace university teachers are not the result of random deviations. They must recognize the statistically proven fact that there are students who are confident that AI will replace university teachers in five years. At least this scientific fact applies to respondents from four Eastern European countries (Poland, Slovakia, Kazakhstan, and Ukraine). These countries have been selected for maximum diversity.

They must consider this scientifically proven fact to make managerial and pedagogical decisions to improve universities' sustainability.

A detailed analysis of students' opinions in different countries helps to compare students' opinions by individual groups of respondents.

### 4.3. Verification of Statistical Hypotheses for Individual Groups of Respondents

To compare students' opinions along the two lines, let us verify statistical hypotheses for each group of respondents.

Let us sequentially test hypothesis 1 and hypothesis 2 for each group of respondents.

Statistical indicators for testing statistical hypotheses are shown in Table 7.

Table 7 shows the statistical indicators that are needed to verify statistical hypotheses. These are sample size (N), average of the sample ($M_{(x)}$), standard deviation for sample ($\delta_x$), and standard deviation for population ($\delta_{x-1}$).

**Table 7.** Statistical indicators of responses by groups of respondents.

| Group of Respondents | N | $M_{(x)}$ | $\delta_x$ | $\delta_{x-1}$ |
|---|---|---|---|---|
| 1. Kazakhstan | 73 | 31.51 | 46.45 | 46.78 |
| 2. Poland | 56 | 1.79 | 13.24 | 13.36 |
| 3. Poland | 39 | 10.26 | 30.34 | 30.74 |
| 4. Poland | 73 | 2.74 | 16.32 | 16.44 |
| 5. Poland | 60 | 3.33 | 17.95 | 18.10 |
| 6. Slovakia | 60 | 1.67 | 12.80 | 12.91 |
| 7. Ukraine | 118 | 15.25 | 35.95 | 36.11 |
| 8. Ukraine | 120 | 11.67 | 32.10 | 32.24 |

*4.4. Verification of Hypothesis 1 for Individual Groups of Respondents*

Let us start by testing Hypothesis 1.

Verification of statistical hypotheses was performed for each group of respondents in Table 8.

**Table 8.** Verification of statistical hypotheses for individual groups of respondents (all students are sure that AI will replace university teachers, $\mu_0 = 100.00$).

| Statistical Indicators | Value for Respondent Groups: | | | | | | | |
|---|---|---|---|---|---|---|---|---|
| | 1 | 2 | 3 | 4 | 5 | 6 | 7 | 8 |
| Sample size, $N$ | 73 | 56 | 39 | 73 | 60 | 60 | 118 | 120 |
| Average of the sample, $M_{(x)}$ | 31.51 | 1.79 | 10.26 | 2.74 | 3.33 | 1.67 | 15.25 | 11.67 |
| Standard deviation for sample, $\delta_x$ | 46.45 | 13.24 | 30.34 | 16.32 | 17.95 | 12.80 | 35.95 | 32.10 |
| Average error, $\dot{S}_{\bar{X}} = \delta_x / \sqrt{n}$ | 5.44 | 1.77 | 4.86 | 1.91 | 2.32 | 1.65 | 3.31 | 2.93 |
| Value $\mid t_{stat} \mid$ for $\mu_0 = 100.00\%$, $(M_{(x)} - \mu_0)/\dot{S}_{\bar{X}}$ | 12.59 | 55.49 | 18.47 | 50.92 | 41.67 | 59.59 | 25.61 | 30.15 |
| Value $t_{tabl}$ for the standard testing level of $\alpha$ (0.05) | 1.645 | 1.645 | 1.686 | 1.645 | 1.645 | 1.645 | 1.645 | 1.645 |
| $\mid t_{stat} \mid > t_{tabl}$ | Yes | Yes | Yes | Yes | Yes | Yes | Yes | Yes |

In Table 8, t-statistics $\mid t_{stat} \mid$ is larger than the $t_{tabl}$ for the given number (100.00%) for each group. So, the authors have accepted the Alternative Hypothesis 1: Not all students are sure that AI will replace university teachers in five years if random deviations are not considered. The result was obtained with a standard hypothesis testing level that equals 0.05.

*4.5. Verification of Hypothesis 2 for Individual Groups of Respondents*

Let us go back to Hypothesis 2.

Verification of statistical hypotheses was performed for each group of respondents in Table 9.

Table 9 shows a very interesting situation. For groups 1, 3, 7, and 8, t-statistics $\mid t_{stat} \mid$ is larger than the $t_{tabl}$ for the given number (0.00%). So, the authors have accepted Alternative Hypothesis 2: Some students are sure that AI will replace university teachers in five years if random deviations are not considered.

For groups 2 and 4–6, t-statistics $\mid t_{stat} \mid$ is less than the $t_{tabl}$ for the given number (0.00%). This means that they have no reason to reject the research hypothesis and accept an alternative hypothesis (Table 9). Therefore, for groups 2 and 4–6, the authors have accepted the Research Hypothesis 2: No student is sure that AI will replace university teachers in five years, if random deviations are not considered.

The result was obtained with a standard hypothesis testing level that equals 0.05.

To make managerial and pedagogical decisions to improve the sustainability of universities, they must consider the above-mentioned scientific results.

**Table 9.** Verification of statistical hypotheses for individual groups of respondents (there are no students who are sure that AI will replace university teachers, $\mu_0 = 0.00$).

| Statistical Indicators | Value for Respondent Groups: | | | | | | | |
|---|---|---|---|---|---|---|---|---|
| | 1 | 2 | 3 | 4 | 5 | 6 | 7 | 8 |
| Sample size, $N$ | 73 | 56 | 39 | 73 | 60 | 60 | 118 | 120 |
| Average of the sample, $M_{(x)}$ | 31.51 | 1.79 | 10.26 | 2.74 | 3.33 | 1.67 | 15.25 | 11.67 |
| Standard deviation for sample, $\delta_x$ | 46.45 | 13.24 | 30.34 | 16.32 | 17.95 | 12.80 | 35.95 | 32.10 |
| Average error, $\dot{S}_{\dot{X}} = \delta_x/\sqrt{n}$ | 5.44 | 1.77 | 4.86 | 1.91 | 2.32 | 1.65 | 3.31 | 2.93 |
| Value $\mid t_{stat} \mid$ for $\mu_0 = 0.00\%$, $(M_{(x)} - \mu_0)/\dot{S}\dot{X}$ | 5.79 | 1.01 | 2.11 | 1.43 | 1.44 | 1.01 | 4.61 | 3.98 |
| Value $t_{tabl}$ for the standard testing level of $\alpha$ (0.05) | 1.645 | 1.645 | 1.686 | 1.645 | 1.645 | 1.645 | 1.645 | 1.645 |
| $\mid t_{stat} \mid > t_{tabl}$ | Yes | No | Yes | No | No | No | Yes | Yes |

## 5. Discussion

AI is a rapidly evolving family of technologies that can bring many economic and social benefits across a full range of industries and social activities [46]. By improving forecasting, optimizing operations and resource allocation, and personalizing service delivery, artificial intelligence can drive socially and environmentally beneficial outcomes and provide critical competitive advantages to companies and the European economy [46]. The areas proposed by the Commission in the Artificial Intelligence Act [46] include education and training [47]. These publications show serious interest in using artificial intelligence in general and education.

Students' attitude towards using artificial intelligence at European universities is also the subject of study. In particular, the authors of [48] studied business students' perceptions of their universities in the Netherlands as they prepared them for an AI work environment; 95 students from 27 universities in the Netherlands took part in the questionnaire survey. The findings indicated that these students believe that their institutions are not currently optimally equipped and/or are not optimally utilizing their capabilities to prepare them for the AI work environment adequately.

The authors of the following work [49] studied students' attitudes towards e-books using AI. The study was conducted at Stockholm University.

A study [50] examined the extent to which German psychology students currently accept and use AI and what influences their acceptance and use. The study was carried out on a sample of 218 psychology students.

The paper [51] examines the expectations of 62 Hungarian university non-technical students. Although respondents were aware of the revolutionary nature of the likely changes, they expressed skepticism about the scale of change.

Unfortunately, the study of student opinion in Romania [1] did not address artificial intelligence.

All of the above sources indicate the relevance of our research. In this study, 599 students of non-information majors were surveyed. They were aware of the revolutionary nature of the likely changes to varying degrees. For the first time, the authors obtained a general picture regarding students' confidence that AI will replace university teachers in five years (Tables 5 and 6).

The study showed that some students in Eastern European countries are confident that AI will replace university teachers in five years. The average number of such students exceeded 10.00%. This result could also be called "skepticism" [51]. However, the results of random deviations cannot explain this size (10.85%). These students have reason to believe that AI will replace university teachers in five years. Such students are seen in the universities of Eastern Europe. This is a weak segment of the market for higher education services [43]. The share of such students may be higher in the strong segments of the higher education market. The opinions of students from strong higher education market segments can be a topic for further research.

Scientific results can be interpreted in the broadest context. It can be seen that there are students in the global student practice who see a threat to the sustainability of higher education. If a wholesale replacement of university teachers with AI technologies occurs, this will disrupt the sustainability of higher education globally.

Scientific results can be analyzed along two comparative lines. The first comparative line consists of four groups of students from the 1st stage (undergraduate study) from four countries. It makes up 1, 2, 6, and 7 groups of students.

The second comparative line consists of four groups of respondents from Poland. These are students of the 1st stage, 2nd stage, students of ordinary postgraduate courses, and students of the Master of Business Administration (MBA) Program. This makes up 2–5 groups of students.

In the first comparative line, among students in the 1st stage (undergraduate study) from Kazakhstan and Ukraine, some students are sure that AI will replace university teachers in five years (Tables 8 and 9). Some students admit the possibility of mass layoffs of university teachers due to the replacement with AI technologies. This fact includes students from countries outside the European Union (EU). In other words, the study shows that some students from Eastern Europe unconsciously allow violation of the sustainability of higher education.

In Poland and Slovakia, the opinions of 1st stage (undergraduate study) students who are sure that AI will replace university teachers in five years can be ignored. It was statistically proven that the EU students surveyed had no reason to see a threat to the sustainability of higher education through the mass layoffs of university teachers.

In the second comparison line, Polish students from different levels of education showed different levels of confidence that AI will replace university teachers in five years. The share of students in the 1st stage (undergraduate study), students of ordinary post-graduate courses, and students of the Master of Business Administration Program can be considered statistically indistinguishable from zero (Table 9). However, some students of the 2nd stage (master study) are sure that AI will replace university teachers in five years. The Ukrainian students of the 2nd stage confirm the opinion of the Polish students of the 2nd stage. Some students of the 2nd stage also admit the possibility of mass layoffs of university teachers due to the replacement with AI technologies. Thus, the analysis of each group of respondents shows a different degree of confidence in students studying at different levels.

The goal of the next stage of the study may be to detail the influence of gender and age differences on students' opinions. It may also be interesting to study the opinions of students of the 3rd stage (PhD) and those associated with IT technologies. Another group of potential respondents are students from Western Europe.

If a wholesale replacement of university teachers with AI technologies occurs, this will disrupt the sustainability of higher education globally. It is not possible to consider this a "smart approach" to teaching and learning based on smart devices and applications [52]. To make managerial and pedagogical decisions to improve the sustainability of universities, the new scientific results obtained above must be considered.

The authors interviewed 599 respondents selected based on maximum diversity. For example, the authors of article [53] interviewed 397 respondents. In [54], the authors decided based on a survey of 77 respondents. Other authors [55] obtained reliable results by interviewing 142 respondents. In the above-mentioned work [51], 62 students were interviewed. Therefore, the surveyed number of respondents, combined with the standard level of statistical hypothesis testing, is sufficient to obtain reliable scientific results.

However, this study has several limitations. First, it is beyond the scope of this article to examine the influence of gender, age, and prior knowledge of AI in the context of our study. Further, the authors surveyed respondents only at universities from four Eastern European countries. Undoubtedly, it would be interesting to continue the research at universities in Western Europe and countries in Africa, Asia, Australia, North and South America.

## 6. Conclusions

The authors explored two divergent hypotheses Hypothesis 1 and Hypothesis 2.

The study showed that some students in Eastern European countries are confident that AI will replace university teachers in five years. The average number of such students exceeded 10.00%. In the first comparative line, among students in the 1st stage (undergraduate study) from Kazakhstan and Ukraine, some students were sure that AI would replace university teachers in five years. In the second comparison line, some Polish students of the 2nd stage (master study) were sure that AI would replace university teachers in five years.

The obtained results allow us to formulate some multilevel managerial and pedagogical recommendations in order to prevent the hidden conflict of SDG 4 and SDG 8 with the consistent introduction of AI technologies in higher education. Implementing these managerial and pedagogical recommendations will increase the sustainability of higher education with the consistent introduction of AI technologies in universities. These recommendations are for university leaders, governments and politicians, researchers and university teachers:

1. University leaders should organize the professional development of university teachers everywhere in the direction of using AI to improve the quality of educational services
2. Governments and politicians are encouraged to enact laws to limit the uncontrolled intervention of AI in higher education. At the same time, it is recommended to take preventive measures for the social protection of university teachers and other teaching staff
3. Researchers are encouraged to continue and expand the study of the opinions of students and university teachers regarding various aspects of the use of AI in higher education. It is important to know in advance how society will benefit and what will have to be sacrificed if AI replaces university teachers
4. University teachers are encouraged to conduct explanatory work among students about the benefits of AI for improving the quality of educational services

The objectives of future research could be as follows:

- to study the influence of gender on the perceptions of Eastern European students on the research topic
- to study the influence of age on the perceptions of Eastern European students on the research topic
- to study the influence of previous knowledge of artificial intelligence on the perceptions of Eastern European students on the research topic
- compare the perceptions of Eastern European students with the perceptions of students from other regions

**Author Contributions:** Conceptualization, V.O.-K., A.A. and W.C.; methodology, V.O.-K., W.C. and Ł.S.; software, V.O.-K., A.A., O.D. and N.A.; validation, V.O.-K., W.C. and A.A.; formal analysis, V.O.-K., A.A., W.C. and N.A.; investigation, V.O.-K., O.D. and A.A.; resources, N.A., W.C. and O.D.; data curation, V.O.-K.; writing—original draft preparation, V.O.-K., A.A., N.A., W.C. and Ł.S.; writing—review and editing, V.O.-K., O.D., Ł.S. and W.C.; visualization, V.O.-K. and A.A.; supervision, V.O.-K. and Ł.S.; project administration, V.O.-K., W.C. and Ł.S.; funding acquisition, O.D. and Ł.S. All authors have read and agreed to the published version of the manuscript.

**Funding:** This research was funded by the EU NextGenerationEU through the Recovery and Resilience Plan for Slovakia under project No. 09I03-03-V01-00130.

**Institutional Review Board Statement:** Experimental studies in this article do not require approval from Review Board.

**Informed Consent Statement:** Informed consent was obtained from all subjects involved in the study.

**Data Availability Statement:** Data are contained within the article.

**Acknowledgments:** The authors warmly thank the students for their time and honest answers. The authors also thank their colleagues who helped organize the survey of respondents. The authors would like to say special words of gratitude to the reviewers for their advice, which helped to improve significantly the quality of the manuscript.

**Conflicts of Interest:** The authors declare no conflict of interest.

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
