# Peer review of "Sustainability of Higher Education: Study of Student Opinions about the Possibility of Replacing Teachers with AI Technologies"

_sustainability, doi:10.3390/su16010055_

Round 1

Reviewer 1 Report

Comments and Suggestions for Authors

See attached file

Comments on the Quality of English Language

Author Response

Please see the attachment. All changes in the article are in blue

Reviewer 2 Report

Comments and Suggestions for Authors

-You mention terms such as SDG 4 and SDG 8 in your abstract. I would write these out and later use the abbreviated forms. You may have readers who do not know what these terms are. 

-Should you use the term university "instructor" rather than teacher in the abstract?

-"The work aimed to study..." Maybe change to- "This research aimed to examine the perceptions of Eastern European students about the possibility of..." (In the abstract)

-  What is the connection between the bibliometric analysis and the actual empirical data? This is a little confusing. (In the abstract)

- I would refrain from terms such as "statistically proven." The way you have worded this section, also makes it seem like you are referring to other studies and not the present study. 

- When you say countries in the European Union and countries outside of the European Union, are you still referring to Eastern European countries in the EU versus non-EU. The wording is a little confusing. 

- Is Eastern European market the best key term? What about Eastern European universities or Eastern European university students? 

Introduction

- Your introduction discusses Romania, but then proceeds to Brazil and China. I would focus more on Eastern European countries if possible. If there is scant literature, then focus on other countries. 

- The Georgia Institute of Technology part might be confusing. People may think you are referring to the country of Georgia rather than the US state, Georgia. 

-You mention that the EU forecasts AI will radically change the education system. This is very vague. You should elaborate on this, especially since you are focusing on European countries. 

- Your purpose statement needs to be more specific and discuss the students more and the content of the study. 

- You mention the "scientific value lies in new empirical data and new research methodology." In this paragraph, there are many claims made. Where is this information coming from? 

Literary Review

- I would provide a little more background information on the sustainability goals for readers who are not familiar with these, especially outside of Europe. What is the origin of these goals? How are they implemented in education in Eastern Europe? 

Bibliometric Analysis

- Why did you just choose Scopus-indexed articles? 

Experiment Planning

- Kazakhstan is not in Europe. I thought this study pertained to Eastern European countries. 

-Instead of using the term Eastern European, I would just identify the countries early on (Kazakhstan, Poland, Slovakia, and Ukraine). You can also mention that two are in the European Union. 

-Why is there a specific category for MBA students? Does this not fall under MA study? 

General Description of the Questionnaire

- Why is there just one survey question? This is a major weakness of the study. Although this question is important, there is so much more valuable data you could get. Why not include a qualitative component as well to elaborate on responses or other relevant closed-ended survey questions? 

-Why is the gender and age data not in Table 2? 

-Why weren't undergrad and grad student data collected at all of the universities?  

Discussion

- There need to be far more concrete connections to the literature. Additionally, you are studying Asian and European countries. The mention of Eastern Europe is confusing. Additionally, Poland and Slovakia are typically defined as central European countries. 

-There should be more specific implications in the conclusion for the specific countries as well as worldwide. There are many limitations in this study. At least a few could be discussed There are also many directions for future research that can also be discussed. 

Comments on the Quality of English Language

Overall, the English language quality is fine. 

Author Response

(The authors gave the same response as above.)

Reviewer 3 Report

Comments and Suggestions for Authors

The article has got strengths however the reviewer has detected some weakness that are described (see document).

Author Response

(The authors gave the same response as above.)

Reviewer 4 Report

Comments and Suggestions for Authors

This paper examined students’ perceptions of replacing university teachers with  AI technologies. The study performed hypothesis testing in Eastern European universities, surveying 599 students. It was concluded that undergraduates’ perceptions of the EU differed from their counterparts in countries outside the EU. The study enriched the AI education research by providing empirical data on students’ perceptions while a lot of current research focuses on discussion and theory. It is recommended that the study perform an appraisal of validity and relativity.

The paper studied students perception on AI replacing teachers in higher education and made comparisons between different groups. 

The paper can be strengthened by appraising its validity and reliability.  

Author Response

(The authors gave the same response as above.)

Round 2

Reviewer 1 Report

Comments and Suggestions for Authors

see report

Comments on the Quality of English Language

see report

Reviewer 2 Report

Comments and Suggestions for Authors

Thank you for making the requested revisions. 

Author Response

Thank you for the valid comments that allowed us to significantly improve the quality of the article

Reviewer 4 Report

Comments and Suggestions for Authors

The authors have addressed the quality appraisal of the study.

Author Response

(The authors gave the same response as above.)

Round 3

Reviewer 1 Report

Comments and Suggestions for Authors

11 ways we helped people with Parkinson’s  ͏ ͏ ͏ ͏ ͏ ͏ ͏ ͏ ͏ ͏ ͏ ͏ ͏ ͏ ͏ ͏ ͏ ͏ ͏ ͏ ͏ ͏ ͏ ͏ ͏ ͏ ͏ ͏ ͏ ͏ ͏ ͏ ͏ ͏ ͏ ͏ ͏ ͏ ͏ ͏ ͏ ͏ ͏ ͏ ͏ ͏ ͏ ͏ ͏ ͏ ͏ ͏ ͏ ͏ ͏ ͏ ͏ ͏ ͏ ͏ ͏ ͏ ͏ ͏ ͏ ͏ ͏ ͏ ͏ ͏ ͏ ͏ ͏ ͏ ͏ ͏ ͏ ͏ ͏ ͏ ͏ ͏ ͏ ͏ ͏ ͏ ͏ ͏ ͏ ͏ ͏ ͏ ͏ ͏ ͏ ͏ ͏ ͏ ͏ ͏ ͏ ͏ ͏ ͏ ͏ ͏ ͏ ͏ ͏ ͏ ͏ ͏ ͏ ͏ ͏ ͏ ͏ ͏ ͏ ͏ ͏ ͏ ͏ ͏ ͏ ͏ ͏ ͏ ͏ ͏ ͏ ͏ ͏ ͏ ͏ ͏ ͏ ͏ ͏ ͏ ͏ ͏ ͏ ͏ ͏ ͏ ͏ ͏ ͏ ͏ ͏ ͏ ͏ ͏ ͏ ͏ ͏ ͏ ͏ ͏ ͏ ͏ ͏ ͏ ͏ ͏ ͏ ͏ ͏ ͏ ͏ ͏ ͏ ͏ ͏ ͏ ͏ ͏ ͏ ͏ ͏ ͏ ͏ ͏ ͏ ͏ ͏ ͏ ͏ ͏ ͏ ͏ ͏ ͏ ͏ ͏ ͏ ͏ ͏ ͏ ͏ ͏ ͏ ͏ ͏ ͏ ͏ ͏ ͏ ͏ ͏ ͏ ͏ ͏ ͏ ͏ ͏ ͏ ͏ ͏ ͏ ͏ ͏ ͏ ͏ ͏ ͏ ͏ ͏ ͏ ͏ ͏ ͏ ͏ ͏ ͏ ͏ ͏ ͏ ͏ ͏ ͏ ͏ ͏ ͏ ͏ ͏ ͏ ͏ ͏ ͏ ͏ ͏ ͏ ͏ ͏ ͏ ͏ ͏ ͏ ͏ ͏ ͏ ͏ ͏ ͏ ͏ ͏ ͏ ͏ ͏ ͏ ͏ ͏ ͏ ͏ ͏ ͏ ͏ ͏ ͏ ͏ ͏ ͏ ͏ ͏ ͏ ͏

see review report

Comments on the Quality of English Language

 It would be good if the authors could „lighten“ the language and style. The message is clear but the text could be edited by a copy editor.
